# Formation, Microstructure, and Properties of Dissimilar Welded Joint between CrMnFeCoNi and Fe

**DOI:** 10.3390/ma16145187

**Published:** 2023-07-24

**Authors:** Krzysztof Ziewiec, Artur Błachowski, Sławomir Kąc, Aneta Ziewiec

**Affiliations:** 1Institute of Technology, Faculty of Mathematics, Physics and Technical Science, Pedagogical University of Cracow, ul. Podchorążych 2, 30-084 Krakow, Poland; 2AGH University of Science and Technology, Faculty of Geology, Geophysics and Environmental Protection, al. Mickiewicza 30, 30-059 Krakow, Poland; sfblacho@cyf-kr.edu.pl; 3AGH University of Science and Technology, Faculty of Metals Engineering and Industrial Computer Science, al. A. Mickiewicza 30, 30-059 Krakow, Poland; slawomir.kac@agh.edu.pl (S.K.); aziewiec@agh.edu.pl (A.Z.)

**Keywords:** high-entropy alloy, dissimilar welding, IR-imaging, scanning electron microscopy, X-ray diffraction, Mössbauer spectroscopy

## Abstract

This research explores the welding process of a high-entropy CrMnFeCoNi alloy with iron, unraveling the intricate chemical compositions that materialize in distinct regions of the weld joint. A mid-wave infrared thermal camera was deployed to monitor the cooling sequences during welding. A thorough analysis of the metallographic sample from the weld joint, along with measurements taken using a nano-hardness indenter, provided insights into the hardness and Young’s modulus. The element distribution across the weld joint was assessed using a scanning electron microscope equipped with an EDS spectrometer. Advanced techniques such as X-ray diffraction and Mössbauer spectroscopy underscored the prevalence of the martensitic phase within the weld joint, accompanied by the presence of bcc (iron) and fcc phases. In contrast, Young’s modulus in the base metal areas displayed typical values for a high-entropy alloy (202 GPa) and iron (204 GPa). The weld joint material displayed substantial chemical heterogeneity, leading to noticeable concentration gradients of individual elements. The higher hardness noted in the weld (up to 420 HV), when compared to the base metal regions (up to 290 HV for CrMnFeCoNi alloy and approximately 150 HV for iron), can be ascribed to the dominance of the martensitic phase. These findings provide valuable insights for scenarios involving diverse welded joints containing high-entropy alloys, contributing to our understanding of materials engineering.

## 1. Introduction

High-entropy alloys (HEAs) are a relatively new class of materials, characterized by their unique microstructure and exceptional mechanical properties [1,2,3,4,5,6,7]. HEAs are multicomponent alloys containing equimolar or near-equimolar amounts of five or more elements. These alloys are of great interest for both fundamental research and practical applications, including structural and functional materials, coatings, and composites. The high configurational entropy may intend to stabilize the disordered phases, such as random solid solution and amorphous phases. The high-entropy solid solutions are typically FCC alloys such as CrMnFeCoNi, discovered by Cantor’s group [8]. For broader utilization of HEAs, the capability to join similar or dissimilar material systems is increasingly being recognized as crucial. Ongoing extensive research into the Cr-Mn-Fe-Co-Ni system alloys has yet to fully explore all aspects of these materials. Notably, the development of effective joining methods, an integral part of potential structural applications, still requires significant attention. The available information on welding techniques for High-Entropy Alloys (HEAs) and the subsequent effects on the structure and properties of the joints remains limited [9,10,11,12]. One of the main challenges in HEA research is the understanding of the complex mechanisms governing the formation and evolution of their microstructure and properties. Among the different synthesis methods, welding is an attractive option to fabricate bulk HEAs and to create complex structures and functional devices. The study of welded HEAs offers an opportunity to investigate the effect of welding parameters on the microstructure and mechanical properties of these materials [13,14,15,16]. This research is focused on the detailed investigation of the microstructure and mechanical properties of a unique welded joint between the high-entropy alloy CrMnFeCoNi and iron. The alloy, with its low diffusion rates in both liquid and solid states, and its ductility due to its face-centered-cubic (fcc) structure, holds particular significance. We carry out a comprehensive analysis of the joint’s cooling curves, and its chemical and phase composition. This thorough examination provides valuable insights into the welding process, potential chemical changes, and the diversity of the joint’s microstructure—elements that collectively influence the quality and strength of the bond. Furthermore, the measurements of the joint’s nanohardness and Young’s modulus yield important information about its strength and deformation resistance, key parameters for evaluating its performance. This study is pivotal to the welding technology domain, contributing to a fuller understanding and potential enhancement of high-entropy alloys for practical applications. 

In the current scientific literature, there is a significant lack of studies using thermography to explore solidification in dissimilar material welded joints. To address this, our work applies thermography to detail the process of solidification during the cooling of a butt-welded joint between the CrMnFeCoNi alloy and iron. This unique approach allows us to visualize and analyze the formation of the welded joint during post-weld cooling, a critical period for determining the microstructure and properties, such as hardness and Young’s modulus. Additionally, the thermographic data are combined with findings from X-ray diffraction analysis and Mössbauer spectroscopy. This approach offers a more in-depth understanding of crystallization in the weld due to the mixing of two distinct chemical compositions. Continuing and expanding upon earlier research on welding dissimilar materials [17,18], the present study leverages experiences gained from investigating the use of thermography to study crystallization processes and phase transformations in metal alloys [19,20]. This work aims to offer fresh perspectives on welded joint formation analysis and furnish engineers with valuable insights for the production of welded joints between iron or low-alloy steels and high-entropy alloys. Ultimately, the findings are expected to bolster the utilization and efficiency of high-entropy alloys.

## 2. Materials and Methods

The CrMnFeCoNi alloy was obtained by melting pure metals: Mn 99.95%, Cr 99.95%, Fe 99.95%, Ni 99.95%, and Co 99.95% in an arc furnace under an argon atmosphere, using 98% titanium as a getter. After each melting cycle, the ingot was inverted and melted again. In order to achieve a homogeneous melt, the ingot was melted seven times. After melting, it was cold-rolled to a thickness of 2.5 mm. The plate was then annealed at 1000 °C for one hour to allow for recrystallization. Similarly, an arc-melted ingot obtained from 99.95% pure Fe was also rolled and annealed to achieve recrystallization at a temperature of 800 °C for one hour. Subsequently, samples of 15 mm × 20 mm were cut from both plates and a butt joint was made. The welding process was conducted in a chamber filled with 99.999% pure argon, with a welding current of 50 A. The welding process was recorded with an FLIR SC7650 thermal imaging camera. Infrared observation and filming were conducted through a port made of CaF_2_. The acquisition was made on a computer station equipped with a ResearchIR thermal imaging system. Figure 1 presents a diagram of this experimental setup. Based on the obtained infrared film, a cooling curve analysis was performed for various areas of the welded joint. The welded joint was subsequently cut perpendicularly to the weld line along the cutting plane and a metallographic specimen was prepared by mechanical polishing and etching in a 3% nitric acid solution in ethanol. Metallographic samples intended for nanoindenter tests were not etched. They were analyzed using a CSM Instruments nanoscratch Berkovich microhardness tester (Needham, MA, USA) [21]. The measurement parameters were as follows—maximum load: 50 mN; loading and unloading speed: 100 mN/min. The surface of the specimen was analyzed using a JEOL 6610 scanning electron microscope equipped with an EDS analyzer. Observations were carried out in secondary electron image mode at an accelerating voltage of 20 kV and a working distance of 10 mm. Then, material from 6 different locations of the transverse specimen was taken for X-ray diffraction measurements and for Mössbauer spectrum measurements. To obtain the material in the form of chips, a 1 mm diameter diamond grinding wheel was used. XRD measurements of the material from the welded joint were performed using a Panalytical X’pert powder diffractometer equipped with an automatic primary beam diaphragm, secondary beam graphite monochromator, and X’celerator strip detector. The measurements were carried out within a range of 30–120° using CuKα radiation and a constant 10 mm exposure length. The step was set at 0.05°. The XRD analyses were performed using Panalytical HighScore software (PANalytical X’Pert PRO MPD) and the PDF-4 database. ^57^Fe Mössbauer spectroscopy measurements were performed at room temperature in transmission geometry by applying the RENON MsAa-4 spectrometer [22] operated in the round-corner triangular mode and equipped with the LND Kr-filled proportional detector. The He-Ne-laser-based interferometer was used to calibrate a velocity scale. A commercial ^57^Co(Rh) source was used. The spectral isomer (center) shifts IS are reported with respect to the isomer (center) shift of room-temperature α-Fe. The absorbers for Mössbauer measurements were prepared using about 10 mg/cm^2^ of investigated materials. 

## 3. Results and Discussion

Figure 2 shows a sequence of frames from a thermographic video obtained during arc heating (Figure 2a–c) and cooling of the welded joint (Figure 2d–h). The cooling curves of the joint after welding, made for areas 1–5 (Figure 2i), show a varied course depending on the location on the joint. The curve recorded for ROI No. 1, starting from a temperature of 1285 °C, shows a continuous decrease in temperature without clear signs of phase transformations—it only has small temperature fluctuations in the initial cooling period, just after turning off the electric arc, related to slight mechanical vibrations of the weld area caused by the displacement of the tungsten electrode. The cooling curve for area No. 2 has a clear stop at a temperature of 1247 °C after cooling to 1205 °C. This may indicate transformations associated with the crystallization of alloy CrMnFeCoNi. In contrast, the curves for areas No. 3 and 4 clearly have temperature stops at much higher temperatures of 1483 °C and 1474 °C, respectively. The equilibrium solidification temperature for iron is 1538 °C, so in the case of the weld, iron crystallization takes place at an undercooling of 55 °C and 63 °C, respectively. The cooling curve in area No. 5, like curve No. 1, does not have a temperature stop in the range below 1415 °C, and its course indicates continuous cooling, without clear thermal effects for phase transformations up to an apparent temperature of about 722 °C, which may be related to the occurrence of an undercooled α-γ transformation.

Figure 3a shows a butt joint with marked EDS analysis locations in the welded joint area, denoted by numbers 1 to 6, while Figure 2b presents the locations of these DSC analyses. Areas with a homogeneous chemical composition represent the base material. No significant concentration gradients are observed on both the high-entropy alloy CrMnFeCoNi side (location 1) and the Fe side (location 6). These are areas with a uniform chemical composition. In contrast, areas 2–5 exhibit variations in chemical composition. This suggests that during welding, two liquids with different chemical compositions formed from diverse materials—the high-entropy alloy CrMnFeCoNi and iron—were present in these areas. The joint produced after the liquids solidified has a non-uniform chemical composition. The formation of this heterogeneity was likely influenced by the limited time the liquid alloys from the two welded materials spent together, but it is also possible that the low diffusion coefficient in the liquid high-entropy alloy had a significant impact on the limited mixing and homogenization of the composition. It should be noted that within the weld area, zones with intermediate chemical compositions between the CrMnFeCoNi alloy and iron are evident. They are marked as A, B, and C in Figure 4. A martensitic microstructure was observed in zones with compositions similar to that of zone A also across the whole weld (Figure 3). Increased hardness was also recorded in these specific zones (Figure 5e).

Figure 5 shows the changes in chemical composition measured by the EDS method along the weld surface (Figure 5b) and within the joint (Figure 5c) along the yellow dashed line (Figure 5a). Changes in Young’s modulus (Figure 5d) and Vickers hardness number (Figure 5e) are also related to the yellow dashed line (Figure 5a). The chemical composition measured on the surface of the joint, at a distance of −6000 μm to approximately −2500 μm from the axis of the weld, exhibits a constant value of around 20% atomic for each of the five elements comprising the alloy CrMnFeCoNi, i.e., Cr, Mn, Fe, Co, and Ni. This corresponds to the region of un-melted base material of alloy CrMnFeCoNi. However, in the range of distances from the axis of the weld from approximately −2500 μm to 4500 μm, an elevated iron content is observed, ranging from 60% at. to 90% at. in the deep regions of the joint corresponding to the yellow dashed line. On the surface, these fluctuations in composition are within the range of 40% at. to 75% at. Fe. In the latter case, a value of 40% at. Fe is found on the side of the weld near the un-melted base material. This part of the Fe concentration plot can be explained by the presence of regions rich in alloy CrMnFeCoNi adjacent to the iron-rich part of the weld (Figure 3, mapping number 5). In turn, the non-uniform composition of the weld and the lack of homogenization during the liquid state of the weld can be explained by the fact that high-entropy alloys usually have low diffusion coefficients, even in the liquid state [23,24,25]. For a distance of 4500 μm from the center of the weld, the iron content sharply increases, as higher values of iron occur in the un-melted form. On the other hand, as the distance from the base material increases, the content of constituent elements of alloy CrMnFeCoNi decreases (excluding iron).

The Young’s modulus (Figure 5d) for distances from the center of the weld below about −4500 μm oscillates around 202 GPa, whereas above this value up to about 3000 μm, the elasticity modulus value changes in the range from approx. 201 GPa to 204 GPa. For distances from the center of the weld higher than 3000 μm, this value continuously increases to approx. 205 GPa. The hardness (Figure 5e) of the base metal of the alloy CrMnFeCoNi varies from 290 HV to 180 HV for distances from the weld axis from −6000 μm to −3000 μm. On the side of the native material of iron (−4500 μm to 8000 μm), the hardness is around 150 HV. However, in the weld zone (−4500 μm to 3000 μm), this value increases up to 420 HV.

Figure 6 depicts the diffraction patterns acquired from material collected from various locations within the welded joint. The welded joint and the specific locations from which the individual samples were taken (numbered from 1 to 6) are presented in Figure 6a. The diffraction pattern from location 1 displays a sequence of peaks originating from planes of the phase isomorphic with the Fe_50_Mn_20_Ni_30_ phase that has a face-centered-cubic structure. The PDF card number for this phase is 01-071-8288. The diffraction pattern from location 2 also contains a series of peaks from a crystal lattice very similar to Fe_50_Mn_20_Ni_30_. However, it also includes peaks from a crystal lattice of pure iron with a body-centered structure (PDF card number 04-007-9753). For the subsequent sampling locations, i.e., 3, 4, and 5, a small peak can be observed at angles of 2θ of 43.7°, 43.8°, and 43.9°, respectively, with no clear evidence of the presence of peaks from other planes. Nonetheless, it can be assumed that these peaks may also be attributed to the presence of a small quantity of phase with a similar crystal structure to the phase observed in the diffraction pattern from sample no. 1. Finally, the diffraction pattern for sample number 6 has peaks from pure iron, and their positions are consistent with the positions of peaks for pure iron with a body-centered structure (PDF card number 04-007-9753).

^57^Fe Mössbauer spectra for samples obtained by filing about 20 mg of material at 1–6 selected locations from the plate are presented in Figure 7 and relevant spectral parameters are listed in Table 1. The spectra for samples from locations 1 and 2 mainly consist of a pseudo-single line with a small value of quadrupole splitting QS ≈ 0.15 mm/s and isomer shift IS = −0.08 mm/s. Such a single-line spectrum is typical for iron in the FCC phase and a small value of quadrupole splitting indicates a local disturbance generating some electric field gradient through the neighboring Cr, Mn, Co, and Ni atoms. On the other hand, the spectra for samples from locations 5 and 6 mainly consist of the magnetically split six-line pattern with a hyperfine magnetic field of about 33 Tesla and zero isomer shift. Such a sextet spectrum is typical for the BCC iron, whereas the spectra for samples 3 and 4 (the central part of the welded joint) indicate the existence of both phases of iron in these locations, but with a clear predominance of the FCC and BCC phase for samples 3 and 4, respectively. A significant decrease in the value of the hyperfine magnetic field of the BCC phase to about 30–32 Tesla for locations 3 and 4 indicates some content of alloying transition metals (Cr, Mn, Co, Ni) in the iron BCC alloy. An additional explanation for the reduced value of the hyperfine magnetic field may be a partial transformation to the martensite phase [26]. This result is consistent with a significant increase in the hardness of the central part of welded joint. The minority spectral component of all six locations is a quadrupole doublet with QS = 0.8–0.9 mm/s and IS = 0.3–0.4 mm/s. This small-area spectral component with a contribution of about 10% or less may be attributed to some intermetallic compound of iron with the alloying transition metals. In addition, the lack of identification of any intermetallic compound in the XRD results may indicate high structural disorder or even amorphization of this minority phase. 

The results showing the cooling curves of selected areas of the welded joint (Figure 2i) are generally consistent with the observed chemical composition of the weld joint surface (Figure 5b). The cooling curve for ROI No. 2 shows a cooling halt at 1248 °C. This value is essentially similar to those of earlier studies [27,28,29] conducted on the CrMnFeCoNi alloy (1289 °C). The lower value observed in the current study is due to a high cooling rate of the weld and solidification at considerable supercooling. On the other hand, the cooling curves of the weld recorded in the areas of ROIs 3 and 4 are characterized by a higher solidification temperature (1483 °C and 1474 °C, respectively), more closely related to the solidification temperature of iron. Thus, these areas contain iron as the majority element, which is also confirmed by the chemical composition analysis of the joint. From the temperature change curves in the ROIs 1 and 2, it appears that these areas of the welded joint do not undergo transformations in the liquid state, which means they lie outside the weld area. This is confirmed by SEM observations, where the analyzed spots marked with numbers 1 and 6 indicate a virtually homogeneous distribution of elements. On the other hand, in areas 2, 3, 4, and 5, there are large differences in the distribution of elements constituting both joined metals. However, it is worth noting that liquid-state diffusion occurring in the weld does not lead to the homogenization of these areas. These high dynamics in the EDS mapping distribution could be due to lower diffusion coefficients of alloying elements in the high-entropy alloy [23,24,25].

Considering the somewhat heterogeneous distribution of chemical elements in the weld, visible on the EDS maps, i.e., in areas 2 ÷ 5 (Figure 3), and also the pointwise EDS analyses carried out both inside the weld and at the surface, it is natural to observe fluctuations in mechanical properties, such as hardness and Young’s modulus, in these places (Figure 5b–e). It is worth noting that the large changes in iron concentration in the weld observed in this study have little effect on Young’s modulus. On the other hand, Young’s modulus for both iron and alloy CrMnFeCoNi are at a similar level. Therefore, this is consistent with the results of other researchers. For example, in studies [30,31], the Young’s modulus for the Cantor alloy was obtained at 202 GPa, and for pure iron in study [32], the longitudinal elasticity modulus was 206 GPa.

It is worth noting that on the side of the base metal of CrMnFeCoNi, the chemical composition change associated with the melting of the CrMnFeCoNi alloy occurs only at a distance of about 1000 μm from the fusion line. On the other hand, on the side of iron, as a base metal, the surface-related change in chemical composition is observed at a distance of about 1000 μm from the weld face. This may be due to a significantly lower diffusivity in the high-entropy alloy compared to the diffusivity of iron. Observations confirming the lower diffusivity of these types of alloys are also presented in studies [23,24,25].

In general, lower hardness is observed on both sides of the weld in the native material than in the weld itself (Figure 5e). On the CrMnFeCoNi alloy side, the hardness ranges between 240 HV and 300 HV, while the hardest areas of the weld have a hardness value of around 400 HV. At the same time, Mössbauer studies’ results at points corresponding to the weld indicate a significant amount of the BCC phase, in which iron is in proximity to a significant amount of other alloying elements. Given that there is a high cooling rate after welding, it can be expected that the BCC phase is of the martensitic type with increased hardness. The average cooling rate, determined based on thermal imaging data, in the range up to 500 degrees is about 35–75 °C/s. Such conditions make it possible for hard metastable phases to form. However, areas of lower hardness can be observed in the weld area. In one such area, the HV hardness is even at the level of 200 HV (at a distance from the weld axis −1500–−1000 μm). In another area of reduced hardness in the weld (near the distance from the weld axis of about +1000 μm), the hardness is about 270 HV. It is worth noting that near these coordinates, areas of the CrMnFeCoNi alloy undissolved in iron are observed (see Figure 3b—area 3 and area 4). This suggests that the varied hardness in the weld may be due to the heterogeneity of the weld’s chemical composition. In turn, the lack of homogenization of the weld may be the result of low diffusivity [23,24,25] in the liquid state of the high-entropy alloy.

During the cooling phase of the autogenous welding process between the CrMnFeCoNi alloy and pure iron, crystallization takes place. This results in areas with varying chemical compositions solidifying next to each other. It is proposed that this variability, possibly linked with the lower diffusion rates typical for high-entropy systems, could affect the local defect density at interphase boundaries [33,34]. Our study reveals at least three distinct phases within the weld: iron in the bcc structure, the fcc phase corresponding to the high-entropy alloy, and martensite. This diversity in both chemical and phase composition might increase the number of interphase boundaries. We acknowledge the potential role of these boundaries acting as sources or sinks for defects, possibly contributing to the increased hardness observed in the weld [33,34]. The intricate nature of these interactions remains a subject of ongoing investigation.

## 4. Conclusions

1. Analysis of cooling curves depicting the temperature changes in different regions of the welded joint reveals the locations where both base metals melted and the weld zone was formed. Furthermore, temperature arrests associated with solidification allowed for identification of the areas where a liquid phase dominated, formed by the high-entropy alloy CrMnFeCoNi, and where an iron-rich liquid phase prevailed. These observations were also confirmed by microstructure analysis and chemical composition analysis using SEM and EDS.

2. A smooth transition between the chemical composition of the CrMnFeCoNi alloy and iron was not observed in the weld zone—EDS analysis revealed the presence of large concentration gradients of chemical elements. This may be attributed to the low diffusivity of the CrMnFeCoNi alloy.

3. Significant changes in mechanical properties were observed in the weld zone. Both Young’s modulus and hardness exhibited larger oscillations in values compared to the base metals, both on the CrMnFeCoNi alloy side and the iron side. In the case of Young’s modulus, the values oscillated around characteristic values of both welded materials. However, the hardness in the weld zone was significantly higher than the hardness of both the CrMnFeCoNi alloy and iron.

4. Based on Mössbauer studies, there were indications that the significant increase in hardness of the weld zone was caused by the presence of a martensitic phase formed during cooling after welding, aided by increased cooling rates. XRD analysis generally indicated the presence of two phases: an iron-rich bcc phase and an fcc phase rich in Mn, Cr, Fe, Ni, and Co. The martensitic-like phase indicated in the Mössbauer studies may thus be isomorphous with the bcc phase.

## Figures and Tables

**Figure 1 materials-16-05187-f001:**
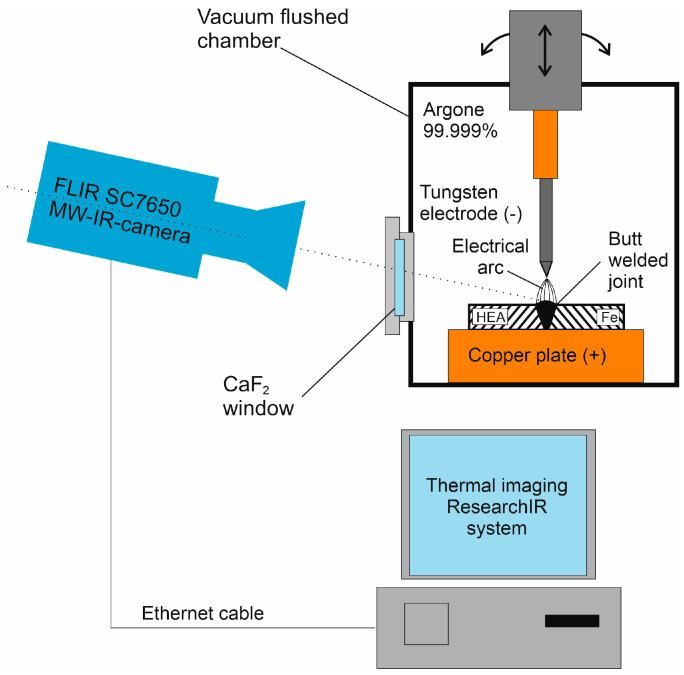
Diagram of the experimental setup, featuring the welding chamber purged with pure argon, FLIR SC7650 thermal camera, and computer station equipped with ResearchIR thermal imaging system (Wilsonville, OR, USA).

**Figure 2 materials-16-05187-f002:**
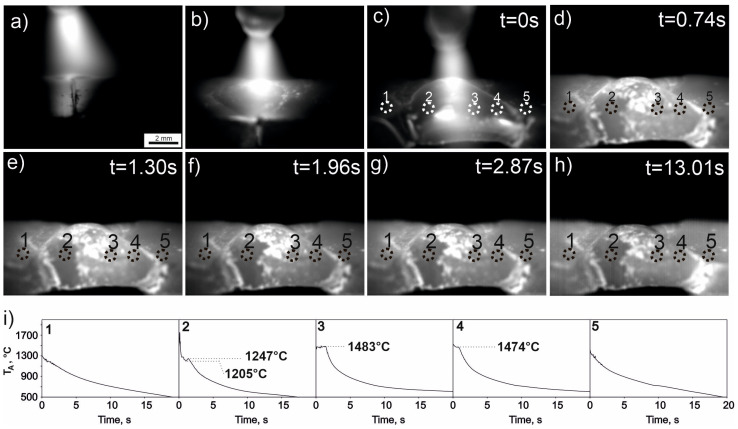
Sequence of thermographic images during the welding process of a butt joint: (**a**) beginning of welding two plates; on the left side is the alloy CrMnFeCoNi and on the right side is Fe; (**b**) formation of a molten metal pool; (**c**) the moment just after completing the weld and before turning off the electric arc, t = 0 s; (**d**) cooling of the weld, a frame from the thermographic video taken at t = 0.74 s after turning off the arc; (**e**) t = 1.30 s after turning off the arc, (**f**) t = 1.96 s after turning off the arc, (**g**) t = 2.87 s after turning off the arc, (**h**) t = 13.01 s after turning off the arc, (**i**) cooling curves after turning off the arc, made based on the thermographic video for the circular areas marked: 1, 2, 3, 4, 5 in images (**c**–**h**); T_A_ stands for apparent temperature.

**Figure 3 materials-16-05187-f003:**
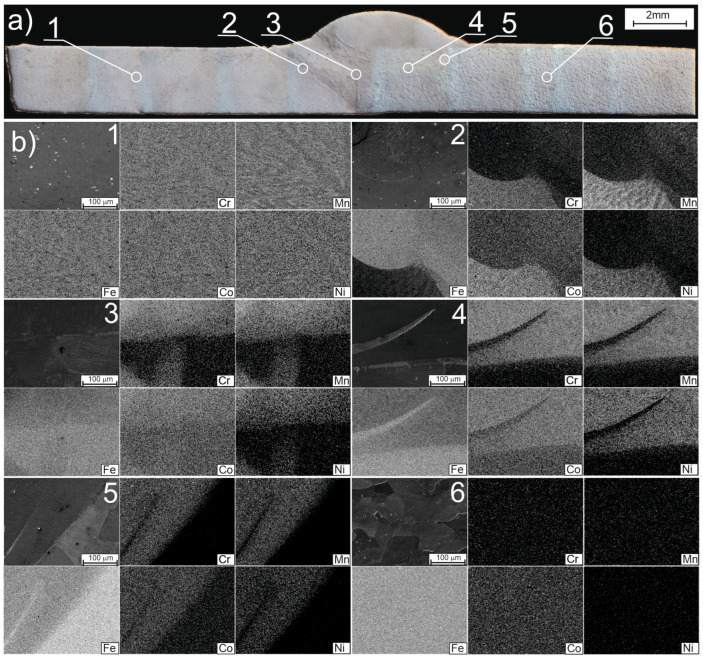
Image of the CrMnFeCoNi-Fe butt joint in a scanning electron microscope: (**a**) unetched smooth surface; (**b**) EDS maps including Cr, Mn, Fe, Co, Ni. Numbers 1 to 6 indicate the locations where the EDS chemical composition map was made.

**Figure 4 materials-16-05187-f004:**
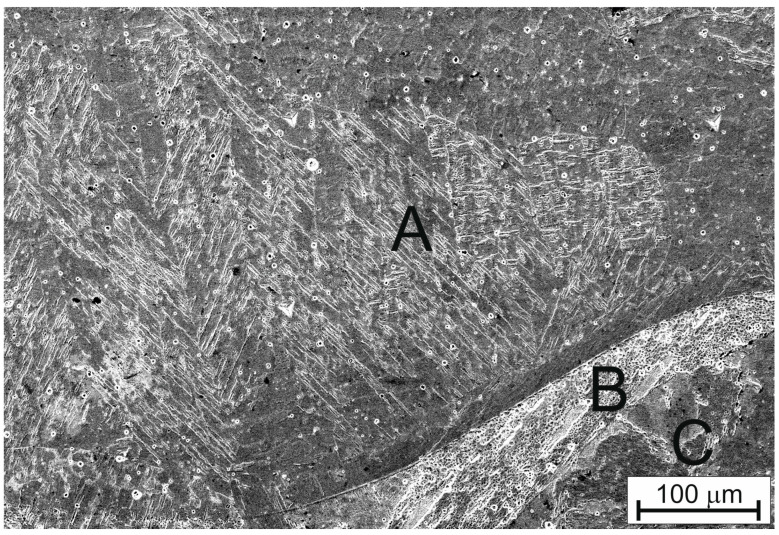
Microstructure from a scanning electron microscope in the area of the weld showing increased hardness; letters A, B, and C indicate the locations of EDS point analyses. A: manganese (Mn)—6.0% at., chromium (Cr)—5.6% at., nickel (Ni)—5.9% at., iron (Fe)—76.0% at., cobalt (Co)—6.6% at.; B: manganese (Mn)—3.6% at., chromium (Cr)—3.1% at., nickel (Ni)—3.5% at., iron (Fe)—86.7% at., cobalt (Co)—3.0% at.; C: manganese (Mn)—2.9% at., chromium (Cr)—3.8% at., nickel (Ni)—3.7% at., iron (Fe)—85.2% at., cobalt (Co)—4.4% at.

**Figure 5 materials-16-05187-f005:**
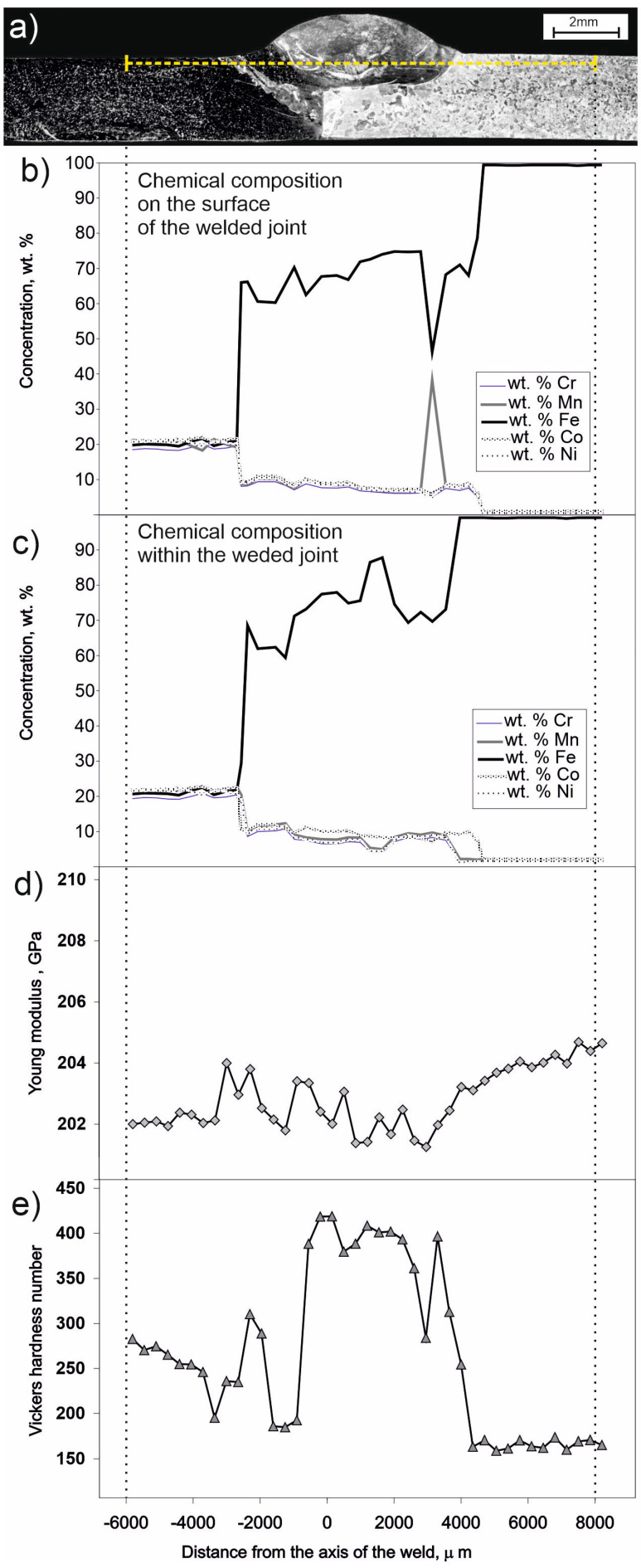
Image of the CrMnFeCoNi-Fe butt joint in a scanning electron microscope: (**a**) etched smooth surface, dashed line represents the location of linear EDS analysis including Cr, Mn, Fe, Co, Ni; (**b**) chemical composition on the surface of the welded joint; (**c**) chart of chemical composition changes determined based on a series of point EDS analyses along the yellow dashed line; (**d**) chart of Young’s modulus changes along the yellow dashed line; (**e**) chart of hardness number HV changes along the yellow dashed line in (**a**). Dotted lines refer to the start and end points of the linear EDS analysis.

**Figure 6 materials-16-05187-f006:**
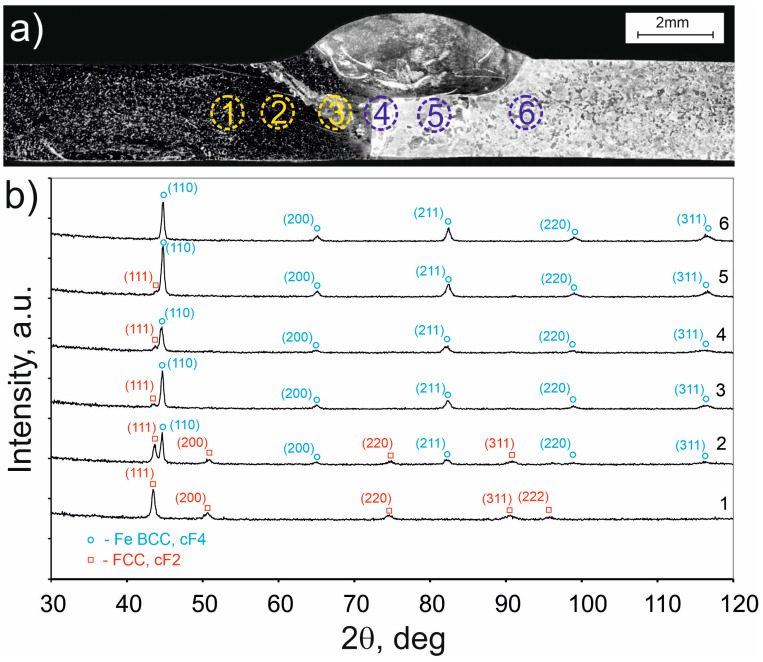
(**a**) Locations from which samples were taken for diffraction studies are marked with numbers 1–6; (**b**) X-ray diffractions from various locations indicated in (**a**).

**Figure 7 materials-16-05187-f007:**
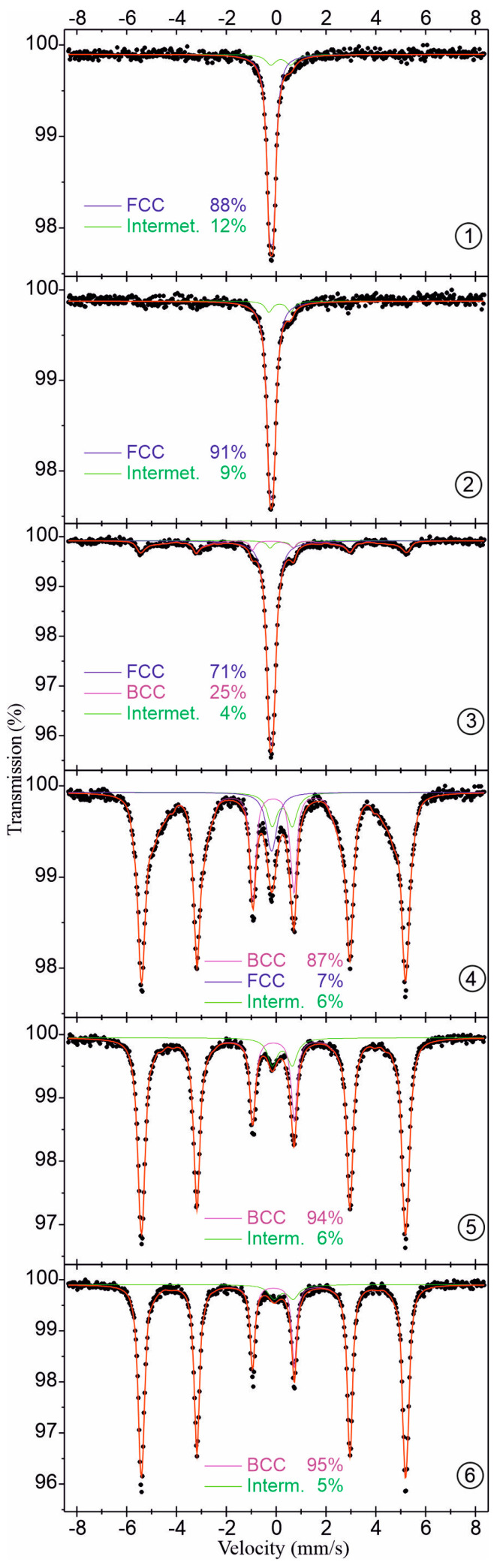
^57^Fe Mössbauer spectra for samples taken from respective locations of the plate. The relative area of the spectral components, corresponding to the percentage distribution of iron atoms onto the FCC/BCC alloys and some intermetallic compound, are shown. For locations 3 and 4, i.e., the central part of the welded joint, the broad spectral component assigned to the BCC phase may partially contain the martensite phase (see text for more details). Numbers 1 to 6 indicate the sampling sites for Mössbauer studies.

**Table 1 materials-16-05187-t001:** ^57^Fe Mössbauer spectroscopy parameters of the spectra shown in Figure 5. Symbol meaning: A—relative area of the spectral components, corresponding to the relative percentage distribution of Fe atoms onto respective phases; IS—isomer (center) shift relative to room temperature α-Fe; QS—quadrupole splitting; B—hyperfine magnetic field; Γ—spectral line-width. Magnetic spectra corresponding to BCC alloy are fitted with the hyperfine magnetic field distribution of the Hesse–Rübartsch method. Hence, the given parameter values for the sub-spectrum of BCC phase correspond to the average values within the hyperfine magnetic field distribution model. Errors are of the order of unity for the last digit shown.

Sample Location	Iron Phase	A (%)	IS (mm/s)	QS (mm/s)	B (Tesla)	Γ (mm/s)
1	FCC	88	−0.08	0.16	-	0.26
	Intermet.	12	0.30	0.80	-	0.50
2	FCC	91	−0.08	0.15	-	0.28
	Intermet.	9	0.30	0.90	-	0.40
3	FCC	71	−0.09	0.15	-	0.30
	BCC	25	−0.04	0.05	30.6	0.35
	Intermet.	4	0.30	0.90	-	0.30
4	FCC	7	−0.08	0.12	-	0.46
	BCC	87	−0.01	0.01	31.9	0.35
	Intermet.	6	0.40	0.80	-	0.50
5	BCC	94	0.00	0.01	32.7	0.25
	Intermet.	6	0.40	0.80	-	0.50
6	BCC	95	0.00	0.01	32.9	0.25
	Intermet.	5	0.40	0.80	-	0.50

## Data Availability

Additional information about the data may be available from the author upon reasonable request.

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
