# Peer review of "Formation, Microstructure, and Properties of Dissimilar Welded Joint between CrMnFeCoNi and Fe"

_materials, 2023, doi:10.3390/ma16145187_

Round 1
Reviewer 1 Report
This paper studied the welding of HEA and Fe. The weld joint has high hardness, and large concentration gradients. The authors thought there be martensite in the welding joint and that's the reason for the high hardness.
However, the evidence for the existence of martensite seems insufficient. Please provide a clear picture of the structure with martensite. otherwise, The high hardness of the weld joint can also be explained by fine grain size or precipitation of IMCs.
In addition, most of the images in the article are blurry, especially Figure 1i, which makes it difficult to see the coordinates clearly.
Also, the article has little in-depth theoretical analysis, thus lacking insufficient innovation.
The overall quality of English language in this article is good. But there are still small mistakes. For example, in line 313, metals is misspelled as metala. Please check carefully.
In addition, the line space of different paragraphs in this article is inconsistent.
Author Response
Reviewer #1
This paper studied the welding of HEA and Fe. The weld joint has high hardness, and large concentration gradients. The authors thought there be martensite in the welding joint and that's the reason for the high hardness.
However, the evidence for the existence of martensite seems insufficient. Please provide a clear picture of the structure with martensite. otherwise, The high hardness of the weld joint can also be explained by fine grain size or precipitation of IMCs.
"We agree with the reviewer's comment and have thus included a clear image of the structure containing martensite. Accordingly, we have made appropriate changes in the article text as follows:
"It should be noted that within the weld area, zones with intermediate chemical compositions between the MnCrNiFeCo alloy and iron are evident. They are marked as A, B, and C in Figure 4. A martensitic microstructure was observed in zones with compositions similar to that of zone A (Figure 4). Increased hardness was also recorded in these specific zones (Figure 5e)."
We have also added a suitable figure to the manuscript, which shows the SEM microstructure of the area with martensite. The remaining figure captions were re-numbered.
In addition, most of the images in the article are blurry, especially Figure 1i, which makes it difficult to see the coordinates clearly.
We agree that Figure 1i is blurry. We have improved the readability of Figure 1i. Regarding the sharpness and resolution of the remaining figures in the pdf file, we do not have control over this as it is determined by the system generating the pdf file. We hope that the final resolution of the photos in the finished article will be better. Just in case, we are sending the original figures and ask for the figures file to be unpacked - they should have good resolution.
Also, the article has little in-depth theoretical analysis, thus lacking insufficient innovation.
We agree with this comment and as a result, the article has been thoroughly re-edited for a more in-depth analysis.
The overall quality of English language in this article is good. But there are still small mistakes. For example, in line 313, metals is misspelled as metala. Please check carefully.
In addition, the line space of different paragraphs in this article is inconsistent.
We agree with this comment. We have corrected the article in such a way that the spacing between lines in the manuscript has been standardized, and typos have been corrected."

Reviewer 2 Report
There are many studies in the literature on welded joints. In this respect, I think the literature summary in the introduction is insufficient. This section can be expanded further with studies from the literature. In addition, at the end of this section, the contribution of the study to the literature and the difference from the literature should be clearly stated.
It is not clear in the paper how the e-modulus values are obtained from the different points in the welded area. Kindly give more details about this.
Author Response
We agree with all the comments of the reviewer and, therefore, we have introduced appropriate corrections and changes to the article. They are presented in detail below, in such a convention that the comments of the reviewer are marked in black, and our explanations and description of the modifications are marked in blue.
Reviewer #2
There are many studies in the literature on welded joints. In this respect, I think the literature summary in the introduction is insufficient. This section can be expanded further with studies from the literature. In addition, at the end of this section, the contribution of the study to the literature and the difference from the literature should be clearly stated.
We agree with the recommendation of the reviewer that there are many works on the welding of high-entropic materials, which is why we provide references to works [13-16] in the introduction. Particularly valuable is the review [15] by Guo, J.; Tang, C.; Rothwell, G.; Li, L.; Wang, Y.-C.; Yang, Q.; Ren, X. Welding of High Entropy Alloys—A Review. Entropy 2019, 21, 431. https://doi.org/10.3390/e21040431. The authors provide a wide overview of works related to the welding of high-entropic alloys. In the revised version of the manuscript, at the end of the Introduction chapter, we write about the differences of the current work in relation to the existing literature on this subject. We also emphasize the importance of current research and we add more references concerning the present work.
It is not clear in the paper how the e-modulus values are obtained from the different points in the welded area. Kindly give more details about this.
This is described in detail in the "Experimental" chapter as follows: „Metallographic samples intended for nanoindenter tests were not etched. They were analyzed using a CSM Instruments nanoscratch Berkovich microhardness tester (Needham, MA, USA). The measurement parameters were as follows - maximum load: 50 mN, load-ing and unloading speed: 100 mN/min. [13] The surface of the specimen was analyzed using a JEOL 6610 scanning electron microscope equipped with an EDS analyzer.”
Similarly, the chemical composition was determined in a scanning electron microscope using an EDS spectrometer. A series of analyzes was performed at each point where nanohardness measurements were made using a Berkovich indenter.
These data are presented in Figure 5.

Reviewer 3 Report
Krzysztof Ziewiec eta al investigates the welding process of a high-entropy MnCrNiFeCo alloy with iron. The process of revealing the complex chemical compositions that form in different regions of the weld joint using experimental studies. The topic is interesting and has industrial applications.
The abstract and introduction are fairly explained. Please introduce a paragraph in the introduction that emphasizes the novelty of the present work. Please highlight the significance of the study.
Please add a schematic illustration of the study.
Pls add quantitative results in abstract.
Figure 1 is not clear , please replot i) 1-4.
How young modulus was determined is not CLEAR?
All Figures are blurry and difficult to comment.
What atomic concentration was selected for the samples and what is effect on the young modulus and hardness changing the concentration ?
XRD Pattern MUST be labelled.
During the alloy composition has any interface effect created? how authors will explain this? interfaces act as a defect for sinks or sources sometimes, did authors see any related phenomena or as the structure becomes complex and it is quite difficult to explain it?
I would suggest adding " Azeem, M. Mustafa, Qingyu Wang, and Muhammad Zubair. "Atomistic simulations of nanoindentation response of irradiation defects in iron." Sains Malaysiana 48.9 (2019): 2029-2039.
Beyerlein, I. J, et al. "Defect-interface interactions." Progress in Materials Science 74 (2015): 125-210.
Author Response
We agree with all the comments of the reviewer and, therefore, we have introduced appropriate corrections and changes to the article. They are presented in detail below, in such a convention that the comments of the reviewer are marked in black, and our explanations and description of the modifications are marked in blue.
These are as follows:
Reviewer #3
Krzysztof Ziewiec eta al investigates the welding process of a high-entropy MnCrNiFeCo alloy with iron. The process of revealing the complex chemical compositions that form in different regions of the weld joint using experimental studies. The topic is interesting and has industrial applications.
"Thank you for this flattering comment; we are pleased that the reviewer appreciated our efforts.
The abstract and introduction are fairly explained. Please introduce a paragraph in the introduction that emphasizes the novelty of the present work. Please highlight the significance of the study.
We agree with the necessity of introducing the suggested changes. We have added the following paragraph:
“In the current scientific literature, there is a significant lack of studies using thermog-raphy to explore solidification in dissimilar material welded joints. To address this, our work applies thermography to detail the process of solidification during the cooling of a butt-welded joint between the MnCrFeNiCo alloy and iron. This unique approach allows us to visualize and analyze the formation of the welded joint during post-weld cooling, a critical period for determining the microstructure and properties, such as hardness and Young's modulus. Additionally, the thermographic data is combined with findings from X-ray diffraction analysis and Mössbauer spectroscopy. This approach offers a more in-depth understanding of crystallization in the weld due to the mixing of two distinct chemical compositions. Continuing and expanding upon earlier research on welding dis-similar materials [17÷18], the present study leverages experiences gained from investigating the use of thermography to study crystallization processes and phase transformations in metal alloys [19÷20]. This work aims to offer fresh perspectives on welded joint for-mation analysis and furnish engineers with valuable insights for the production of welded joints between iron or low-alloy steels and high-entropy alloys. Ultimately, the findings are expected to bolster the utilization and efficiency of high-entropy alloys.”
Please add a schematic illustration of the study.
A schematic illustration of the study has been prepared and added as Figure 1.
Pls add quantitative results in abstract.
Quantitative results were added to the abstract.
Figure 1 is not clear , please replot i) 1-4.
Figure 1 has been revised as recommended by the reviewer. After renumbering it is Figure 2.
How young modulus was determined is not CLEAR?
We agree with this remark and we have corrected the manuscript. The list of literature has been supplemented with this item. A note was introduced in the text: "Young's modulus was determined according to the methodology given in references.
Oliver, W.C., Pharr, G.M., 1992. An improved technique for determining hardness and elastic-modulus using load and displacement sensing indentation experiments. Journal of Materials Research 7 (6), 1564–1583.
All Figures are blurry and difficult to comment.
We agree that the quality of the file is poor, which affects the resolution, but the authors have no influence on it, so we send all drawings with good resolution in the attachment.
What atomic concentration was selected for the samples and what is effect on the young modulus and hardness changing the concentration ?
Cantor's alloy [8] was selected and the chemical composition is MnCrNiFeCo, i.e. Mn - 20 at.%; Ni - 20% at.; Fe - 20% at.; Co - 20% at. therefore, it is a multi-component alloy and in order to determine the full dependence of Young's modulus on the composition, it should be done for 5 changing concentrations of individual alloying elements. – This will be the subject of our future work, due to the huge number of variants to be explored. However, the dependence of hardness and Young's modulus on concentration is shown in Figure 5 in our work. – We attach a file with drawings in good resolution where you can observe these relationships.
XRD Pattern MUST be labelled.
The diffraction pattern was corrected and the lattice planes of the phases were assigned appropriate indices.
During the alloy composition has any interface effect created? how authors will explain this? interfaces act as a defect for sinks or sources sometimes, did authors see any related phenomena or as the structure becomes complex and it is quite difficult to explain it?
I would suggest adding " Azeem, M. Mustafa, Qingyu Wang, and Muhammad Zubair. "Atomistic simulations of nanoindentation response of irradiation defects in iron." Sains Malaysiana 48.9 (2019): 2029-2039.
Beyerlein, I. J, et al. "Defect-interface interactions." Progress in Materials Science 74 (2015): 125-210.
We have added a relevant passage to the manuscript in which we refer to the suggested references. This is the following excerpt:
During the cooling phase of the autogenous welding process between the MnCrNiFeCo alloy and pure iron, crystallization takes place. This results in areas with varying chemi-cal compositions solidifying next to each other. It's proposed that this variability, possibly linked with the lower diffusion rates typical for high-entropy systems, could affect the lo-cal defect density at interphase boundaries [33,34]. Our study reveals at least three distinct phases within the weld: iron in the bcc structure, the fcc phase corresponding to the high-entropy alloy, and martensite. This diversity in both chemical and phase composi-tion might increase the number of interphase boundaries. We acknowledge the potential role of these boundaries acting as sources or sinks for defects, possibly contributing to the increased hardness observed in the weld [33,34]. The intricate nature of these interactions remains a subject of ongoing investigation.

Round 2
Reviewer 1 Report
This paper has been well revised according to the reviewer's advise, and the reviewer thinks it can be published.
Author Response
Dear Reviewer,
Thank you for taking the time to review my work. I greatly appreciate your insights and feedback. Your contributions have been invaluable and will certainly enhance the quality of my work.
Best regards,
Krzysztof Ziewiec,
corresponding author